# The gold content of mafic to felsic potassic magmas

Jia Chang [1] ✉, Andreas Audétat [1] & Thomas Pettke [2]

Many epithermal gold and gold-rich porphyry-type ore deposits are associated with potassic magmas. Hence, potassic magmas are commonly assumed to have been unusually Au-rich or to have contained high Au/Cu ratios. However, these hypotheses remain poorly tested. Here, we report Au concentrations and Au/Cu ratios in silicate melt inclusions analyzed in potassic rocks worldwide. The results suggest that mafic potassic magmas generally contain only 2–7 ng/g Au, despite common sulfide exhaustion during partial mantle melting. Both the absolute Au concentrations and Au/Cu ratios are comparable to those of mafic calc-alkaline magmas, and they vary little during subsequent magma differentiation because magmatic sulfide precipitation is strongly dominated by monosulfide solid solution that is relatively poor in Au and Cu. We thus suggest that the close association of Au-rich deposits with potassic magmas is not due to Au enrichment in the magma, but rather due to selective Au precipitation at the hydrothermal stage.

Epithermal Au and Au-rich porphyry-type ore deposits tend to be genetically associated with potassic magmas. About 40% of the largest Au-rich deposits are related to potassic igneous rocks, which comprise only about 5–10 vol% of the global arc igneous rocks[1,2]. Therefore, it is commonly assumed that potassic magmas are exceptionally Au-rich[2–5]. In some studies, this has been attributed to unusually Au-rich mantle sources[3,5,6], whereas other studies propose that Au became enriched in potassic magmas during the process of partial mantle melting[4,7–9] or during subsequent magma differentiation[10,11]. Some recent studies also argued that high magmatic Au/Cu ratios, rather than high absolute Au concentrations, render potassic magmas productive for Au-rich porphyry-type Cu deposits[12,13].

Enrichment of Au in magmatic-hydrothermal ore deposits may also be explained by non-magmatic factors. One option is indicated by an experimental study showing that the partitioning of Au into magmatic fluids is amplified in the presence of high alkali contents[14]. Alternatively, the commonly extensional tectonic setting in which potassic magmas are generated[3,15] may promote the development of shallowly emplaced, little evolved magma reservoirs, which, in turn, may affect the hydrothermal evolution and lead to selective Au precipitation in the deposits[16,17].

The hypothesis of magmatic Au enrichment can be tested by comparing the metal content of potassic magmas with that of calc-alkaline magmas. Available evidence to date is largely based on bulk-rock analysis of potassic porphyry dikes or subaerial volcanic lavas[12,18–21]. However, since these rocks cooled relatively slowly after magma emplacement, significant loss of Au may have occurred during magma degassing[22]. The Au content of bulk rocks can also get modified during subsequent hydrothermal alteration[19,21]. Melt inclusions, i.e., droplets of silicate melt trapped in phenocrysts during magma crystallization, provide a way to minimize these problems because they are little affected by magma degassing or subsequent hydrothermal alteration. However, the Au content of melt inclusions is representative of bulk magmas only if the magmas did not yet reach sulfide saturation, as magmatic sulfides strongly scavenge chalcophile elements in magmas[23,24]. Data on Au contents of melt inclusions are rare[10,25–27], mostly because it is very challenging to analyze Au concentrations at parts per billion levels in samples weighing only a few tenths of a microgram.

In the present study, we analyzed Au and Cu concentrations in melt inclusions from mafic to felsic potassic rocks worldwide, using state-of-the-art laser-ablation inductively-coupled-plasma mass-spectrometry (LA-ICP-MS) (Methods). To explain the measured Au

[1]Bavarian Geoinstitute, University of Bayreuth, Bayreuth 95440, Germany. [2]Institute of Geological Sciences, University of Bern, Bern 3012, Switzerland. ✉e-mail: Jia.Chang@uni-bayreuth.de

concentrations and Au/Cu ratios in potassic melts, we additionally performed quantitative modeling based on (1) the abundance and composition of sulfides during mantle partial melting and subsequent magma fractionation and (2) results of high pressure-temperature experiments.

## Results and discussion

### Au contents and Au/Cu ratios of mafic potassic melts

We analyzed entire, unexposed melt inclusions hosted in olivine ($n = 94$) and clinopyroxene ($n = 27$) from mafic potassic dikes and volcanic rocks collected from 11 locations worldwide (Figs. 1 and 2). Our petrographic studies suggest that olivine in the potassic mafic rocks was the liquidus phase, followed by crystallization of clinopyroxene. Careful inspection of more than 100 polished thick sections revealed that olivine phenocrysts are mostly free of sulfide inclusions, whereas clinopyroxene phenocrysts commonly do contain sulfide inclusions, especially in clinopyroxene-rich samples (Supplementary Data 1). Therefore, the potassic magmas were generally sulfide-undersaturated during the crystallization of olivine, but reached sulfide-saturation during the crystallization of clinopyroxene.

Because chalcophile elements (e.g., Au and Cu) partition strongly into sulfide phases[23,24] but barely into silicate minerals and Fe-Ti oxides[28], the content of these metals in olivine-hosted melt inclusions prior to sulfide saturation should be representative of the bulk magma. Measured Au concentrations in olivine-hosted melt inclusions range from 1.5 to 6.5 ng/g Au (average 3.2 ng/g Au; Supplementary Data 2), which is similar to Au contents reported from mafic calc-alkaline

magmas (0.6–5.5 ng/g Au; average 2.6 ng/g; Fig. 3a). Most of the mafic calc-alkaline magmas were sulfide-undersaturated[27,29], too. This suggests that Au contents of mafic potassic and mafic calc-alkaline magmas are similar. The same is true for the Au/Cu ratios (Fig. 3b). The relatively low Au/Cu ratios of the melt inclusions analyzed from Wozhong, Two Buttes, and West Eifel can be explained by post-entrapment gain of Cu[30,31], as these melt inclusions are coarsely crystallized and thus resided for prolonged times at high temperatures. The original Cu content of silicate melts at Wozhong, Two Buttes, and West Eifel can be constrained based on the Cu content of the most Cu-rich sulfide inclusions (i.e., rare sulfide liquid or intermediate solid solution, the first sulfides that precipitate when sulfide saturation was reached) and experimentally determined sulfide–silicate melt partition coefficients (see Supplementary Data 3). Correspondingly corrected Au/Cu ratios in the silicate melts are similar to those of the other mafic potassic magmas investigated, and also similar to those of mafic calc-alkaline magmas (Fig. 3b). Both the absolute Au contents and Au/Cu ratios of mafic potassic and mafic calc-alkaline magmas are slightly higher on average than those of mid-ocean ridge basalts (Fig. 3a, b).

The melt inclusion data also suggest that mafic potassic magmas tend to be more S-rich and more oxidized than mafic calc-alkaline magmas (Fig. 3c, d), as noticed previously[32–35]. Importantly, the S contents of both magma types lie mostly below the sulfur solubility curve at the P-T conditions prevailing during mantle partial melting (Fig. 4), which suggests that the sulfides in the mantle source became usually exhausted during the partial melting process. The Au and Cu contents of the primary mantle melts can thus be estimated

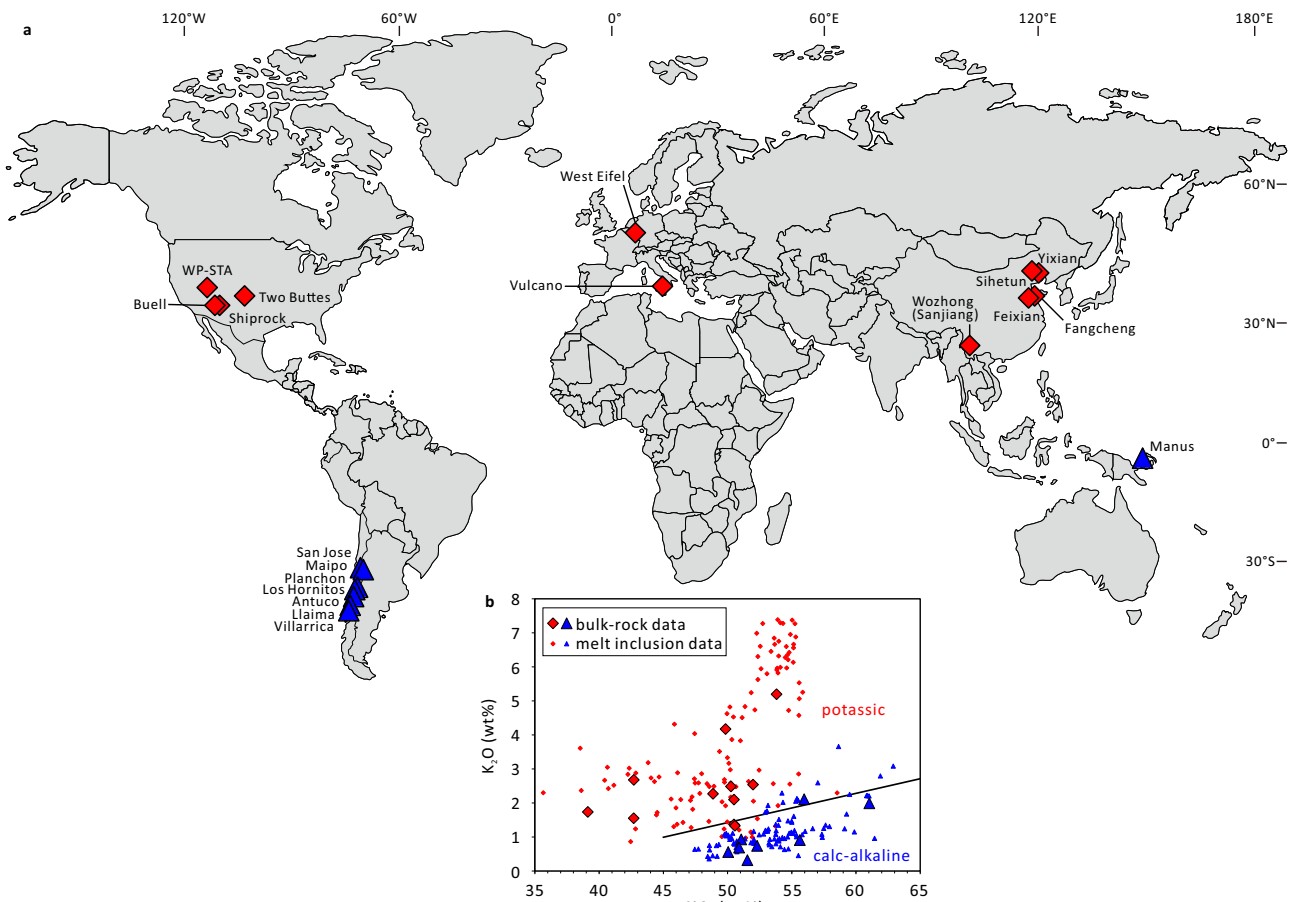

**Fig. 1 | Origin of samples discussed in the present study. a** Distribution of samples. The base map is modified after https://commons.wikimedia.org/wiki/File: Simplified_World_Map.svg. **b** K$_2$O vs SiO$_2$ contents of melt inclusions and corresponding bulk rocks (the bulk-rock compositions were taken from references listed in Supplementary Data 1). The line that separates potassic rocks from calc-alkaline rocks is from Rickwood[68]. Mafic potassic samples are shown in red diamonds (this study), whereas calc-alkaline arc basalts with previously published Au concentrations in melt inclusions are shown in blue triangles[25,27,29].

independently if one knows the abundance and metal content of mantle sulfides, and the degree of mantle partial melting. Since primitive potassic melts typically form by partial melting of phlogopite ± amphibole-veined peridotites[36–40, this study], we studied 29 natural, veined peridotites mostly from the Eifel region in Germany to constrain the abundance and composition of mantle sulfides (Supplementary Data 5). To quantify the metal content of the sulfides, 70 unexposed sulfide inclusions were drilled out and analyzed as entities by LA-ICP-MS (Supplementary Data 6). The metasomatic veins commonly contain abundant sulfides (up to 1.6 wt%), whereas the surrounding peridotites contain virtually no sulfides (e.g., Fig. 5). High P-T melting experiments of veined peridotites suggest that 20–100 wt% melting of

the vein material with only minor addition of the surrounding peridotite is required to reproduce the composition of natural mafic potassic magmas[36–40, this study] (Supplementary Figs. 1 and 2, and Supplementary Data 7 and 8).

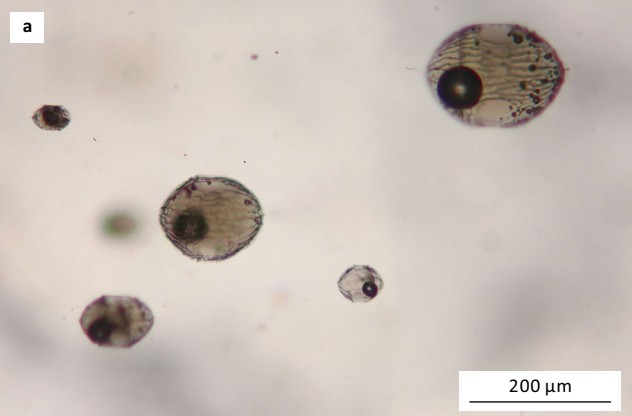

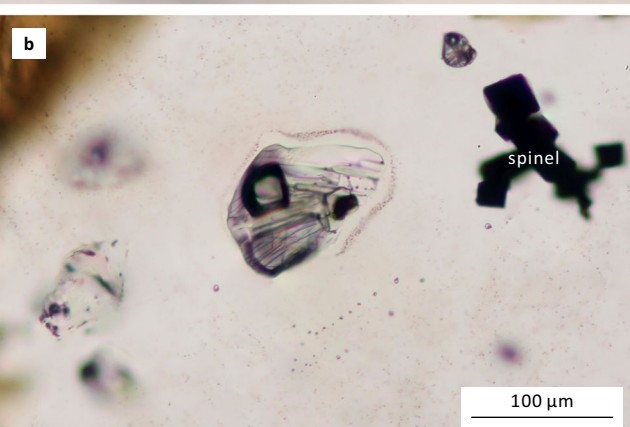

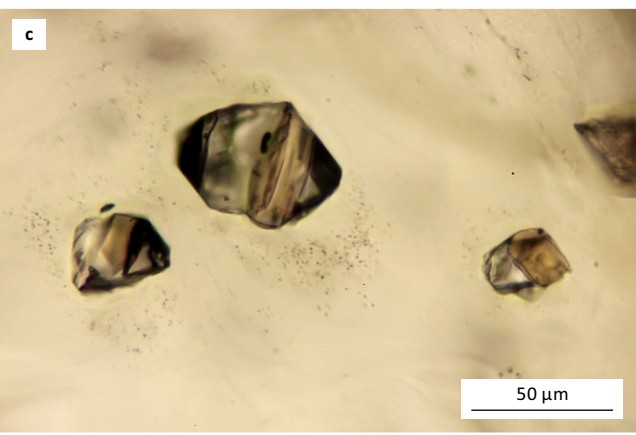

**Fig. 2 | Representative melt inclusions in the studied mafic potassic rocks.** **a** Glassy melt inclusions with shrinkage bubbles hosted in an olivine phenocryst, Vulcano, Italy. **b** Crystallized melt inclusions hosted in an olivine phenocryst, Sihetun, China. Notice the coexisting spinel inclusions. **c** Crystallized melt inclusions hosted in a clinopyroxene phenocryst, Two Buttes, USA. The images were taken under transmitted light and enhanced by focus stacking.

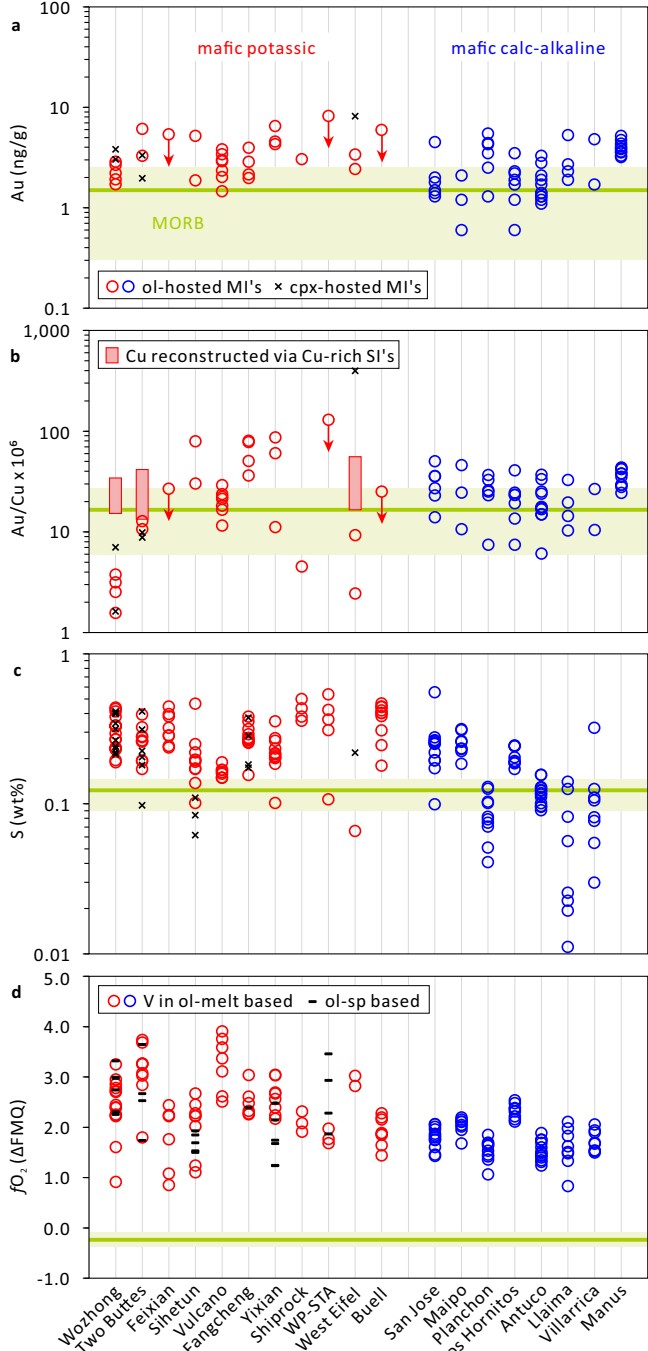

**Fig. 3 | Gold content, Au/Cu ratio, S content, and $fO_2$ of mafic potassic vs mafic calc-alkaline magmas. a–c** The melt inclusion data from potassic magmas are reported in this study, whereas those from calc-alkaline magmas are taken from the literature[25,27,29]. Ranges and averages of MORB (i.e., mid-ocean ridge basalts, in green) are shown for comparison[69]. **d** The estimated oxygen fugacities, which are expressed in log units relative to fayalite-magnetite-quartz (FMQ) buffer, are based on V partitioning between melt inclusions and olivine[60] and/or the composition of spinel inclusions and their olivine host[70] (Supplementary Data 4). MORB data are also shown for comparison[71]. The slightly lower Au contents and Au/Cu ratios of MORB can be explained by the fact that these magmas are generally sulfide-saturated, which is a consequence of their lower $fO_2$ (ref. 70). ol olivine, cpx clinopyroxene, sp spinel, MI melt inclusion, SI sulfide inclusion.

The veined peridotite shown in Fig. 5, for example, would produce a sulfide-undersaturated, primitive potassic melt that contains 1.2–5.8 ng/g Au and a Au/Cu ratio of ~20 × 10⁻⁶. More generally, to explain the 0.07–0.42 wt% S observed in the mafic potassic melts shown in Fig. 4, the metasomatic components would need to have contained 0.036–0.23 wt% to 0.18–1.2 wt% sulfides at melting degrees of 20% to 100%, respectively, given a typical sulfide S content of 36.5 wt%. Together with an observed broad correlation between sulfide

abundance and sulfide Au and Cu contents in metasomatic mantle veins (Supplementary Fig. 3), the Au contents and Au/Cu ratios of the primitive potassic magmas can be estimated at 1.2–6.6 ng/g and $1.3 \times 10^{-6}$–$77 \times 10^{-6}$, respectively. These predicted values agree well with those observed in the mafic melt inclusions (Fig. 3a, b). Consequently, neither the mantle source nor the primitive potassic partial melts are characterized by unusually high Au contents or high Au/Cu ratios.

## Au contents and Au/Cu ratios of evolved potassic melts

To test whether potassic magmas can attain high Au contents or high Au/Cu ratios during magma differentiation, we additionally analyzed 38 melt inclusions in 13 intermediate to felsic potassic rocks in the Sanjiang region of southwestern China (Supplementary Data 9) and complemented the data with quantitative modeling. Previous studies demonstrated that the entire range of mafic to felsic potassic magmas in the Sanjiang region formed by fractionation of mantle-derived mafic potassic melts[41,42]. Our melt inclusion data suggest that the evolved potassic melts did not contain more Au than their mafic parents (Fig. 6a). Moreover, Au concentrations and Au/Cu ratios in the evolved potassic melts are comparable to those in similarly evolved calc-alkaline melts (Fig. 6). Consequently, the Au contents and Au/Cu ratios of evolved potassic magmas are not unusually high either.

The quantitative crystallization models shown in Fig. 6 suggest that Au concentrations and Au/Cu ratios in the evolved potassic and calc-alkaline melts are best captured by dominant MSS fractionation with only very minor sulfide liquid. LA-ICP-MS analysis of 41 unexposed sulfide inclusions in this study suggests that both Cu-rich sulfide liquid and Cu-poor MSS were present during the onset of sulfide saturation in mafic potassic magmas at Sanjiang, Two Buttes and West Eifel (Supplementary Data 3), whereas solely MSS was present at later stages. This is also the case in several other magmatic systems (Santa Rita[43,44], USA; La Fossa[45], Italy; Chilas[46], Kohistan; Mont-Dore Massif[47], France). These observations agree well with previous quantitative modeling[44,48–50] and high P-T experimental studies[41,51], because the precipitation of sulfide liquid rapidly depletes the residual silicate melt in Cu, such that only MSS can precipitate afterwards. MSS-dominated

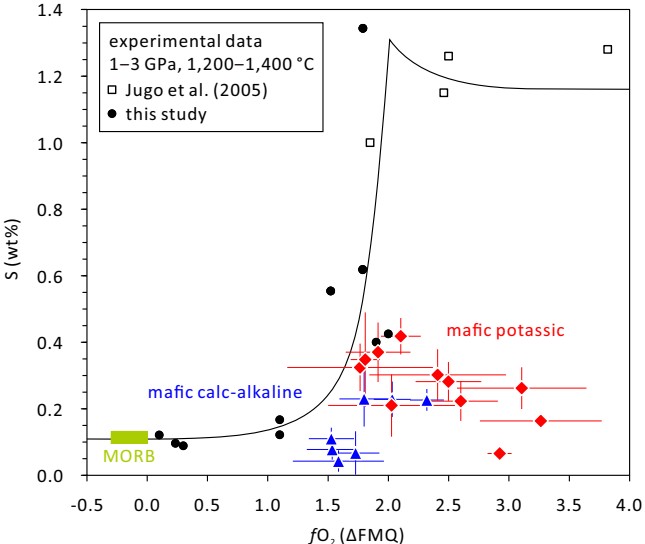

**Fig. 4 | Sulfur content vs $f$O₂ of mafic potassic and mafic calc-alkaline magmas.** The S solubility curve was constructed from experimental data of Jugo et al.[34,72], which were obtained on mafic calc-alkaline melt compositions, and from experimental data obtained in this study on mafic potassic melt compositions. The red diamonds show the sulfur contents and $f$O₂ of natural mafic potassic melt inclusions analyzed in this study, whereas the blue triangles represent mafic calc-alkaline melt inclusions analyzed in a previous study[27]. Error bars denote one standard deviation of multiple analyses. MORB, mid-ocean ridge basalts[69,71].

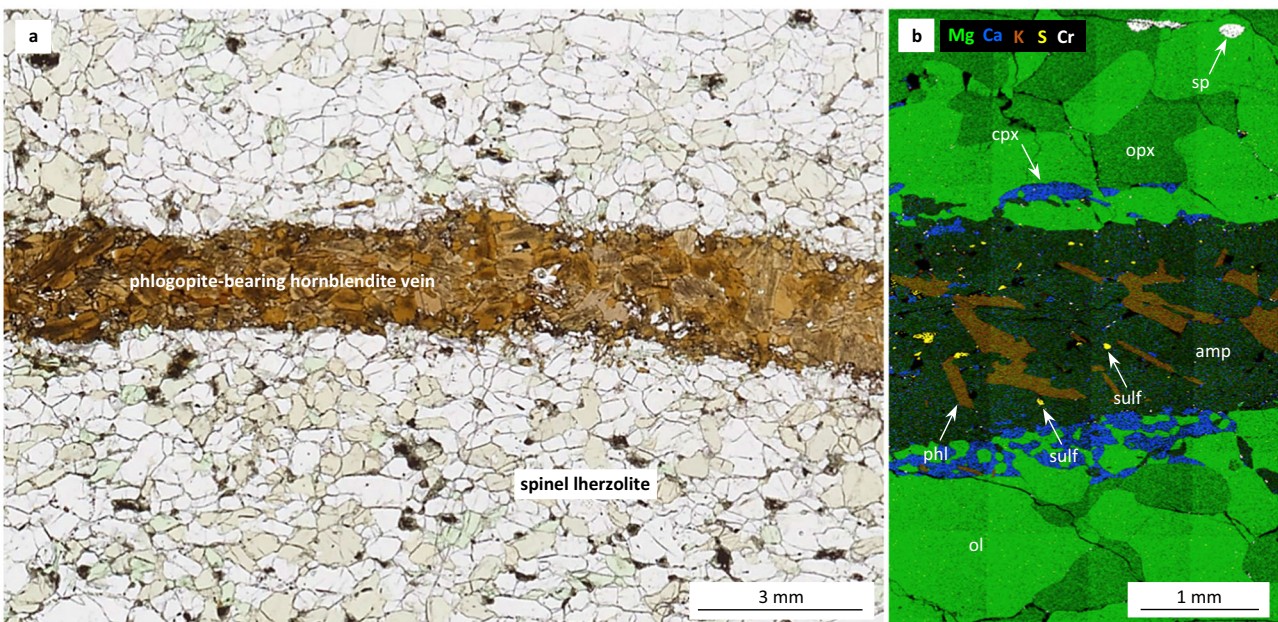

**Fig. 5 | Petrology of a metasomatic vein in peridotite from the West Eifel, Germany. a** Transmitted-light image of a spinel lherzolite cut by a phlogopite-bearing hornblendite vein (ME42). **b** SEM-EDS elemental composite map of part of the same sample. The vein hosts ~0.21 wt% sulfides that contain ~2.8 wt% Cu and

~0.55 μg/g Au (Au/Cu ~20 × 10⁻⁶), whereas the surrounding lherzolite contains no sulfides. amp amphibole, cpx clinopyroxene, ol olivine, opx orthopyroxene, phl phlogopite, sp spinel, sulf sulfide.

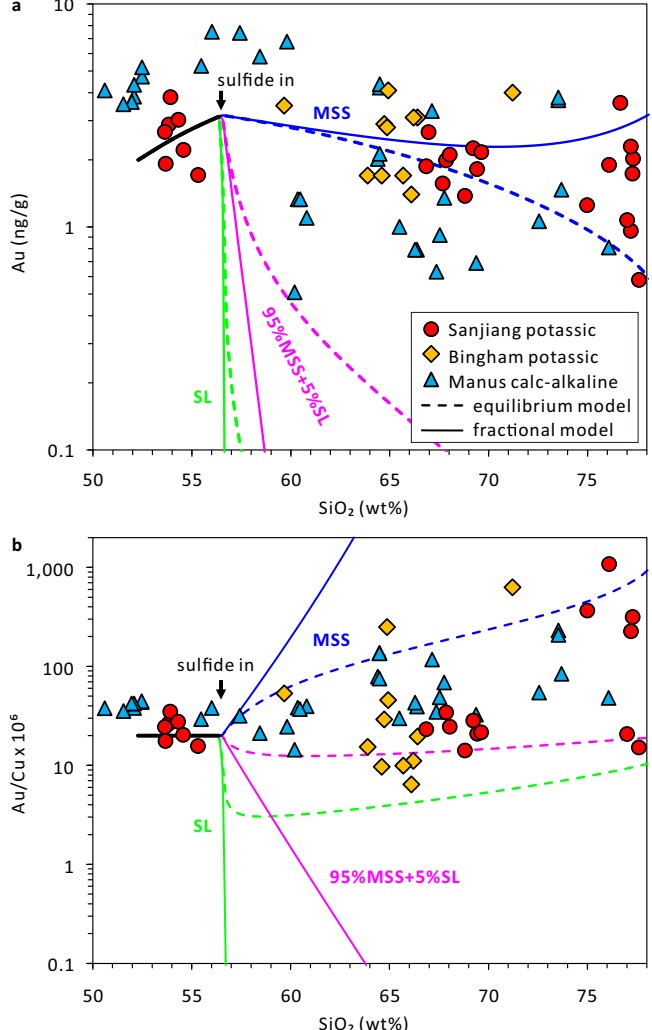

hydrothermal systems. For example, shallowly emplaced porphyry Cu deposits (~1–3 km paleodepths) are generally characterized by high-temperature bornite(-magnetite) ore assemblages and high ore Au/Cu ratios, whereas their deeper equivalents are characterized by lower-temperature chalcopyrite-pyrite ore assemblages and lower ore Au/Cu ratios[16,57]. Experimental data suggest that bornite can incorporate an order of magnitude more Au than coexisting chalcopyrite[58], which provides a viable explanation why shallow porphyry deposits tend to be more Au-rich than deeper ones[16].

## Methods
### Natural samples and petrography
We studied a total of 97 samples of high-Mg mafic potassic volcanic rocks or dikes from 11 locations in China, USA, Germany, and Italy (Fig. 1; Supplementary Data 1), to quantify Au concentrations in mafic potassic melt inclusions and to constrain the timing and extent of magmatic sulfide saturation. From each sample, one or two doubly-polished thick sections were prepared (~3.5 cm × ~2.5 cm; with a thickness of 120–200 μm), except for the scoria sample from Vulcano, for which 10 olivine grains were selected and doubly-polished for melt inclusion analysis. We performed detailed petrographic studies on all the sections, and selected some of them for further LA-ICP-MS analysis and electron probe microanalysis (EPMA).

Additionally, we investigated metasomatic mantle rocks from West Eifel in Germany (24 samples), Shandong Fangcheng in the North China Craton (3 samples), Pyrenees in France (1 sample), and Finero in Italy (1 sample) to quantify the occurrence, abundance and composition of mantle sulfides (Supplementary Data 5). One doubly-polished thick section was prepared from each sample. The abundance of sulfides was determined by comparing the area of exposed sulfides with the total surface area[22,44], and then was transformed into weight fractions using respective densities (Supplementary Data 5). Many sulfide grains in the xenolith samples from West Eifel were altered to spongy Fe-oxides interspersed with minor sulfide remnants. The alteration likely occurred during cooling and degassing of the magmas that transported the xenoliths to shallow levels[59], hence the spongy Fe-oxide blebs were counted as sulfides. A few entire, unexposed sulfide inclusions were analyzed by LA-ICP-MS in each sample.

We also collected 4 intermediate potassic rocks and 9 felsic potassic rocks from the Sanjiang region in southwestern China to quantify the Au contents of evolved potassic melt inclusions. One to six doubly-polished thick sections were prepared from each sample, with the aim of finding large enough and nicely preserved melt inclusions for LA-ICP-MS and EPMA analysis.

Some of our samples collected from the Sanjiang region and those collected from the North China Craton (Sihetun, Yixian, Feixian, and Fangcheng) are genetically associated with the giant Beiya skarn-porphyry Au-Cu deposit[42] and numerous magmatic-hydrothermal lode Au deposits[20], respectively.

### Piston cylinder experiments
To investigate the solubility of S at sulfide saturation in potassic mantle melts as a function of $fO_2$, and to constrain the partial melting degrees that are required to reproduce the composition of natural high-Mg mafic potassic magmas, we conducted 10 partial melting experiments on phlogopite- and sulfide-bearing metasomatized mantle material in end-loaded, solid-media piston cylinder presses hosted at the Bavarian Geoinstitute (BGI), University of Bayreuth, Germany. A friction correction of 0.12 GPa was applied to the utilized MgO-NaCl assemblies based on in-house pressure calibrations, resulting in total pressure uncertainties of around ±0.04 GPa. Run temperatures were monitored by S-type thermocouples (Pt-Pt$_{90}$Rh$_{10}$) with an uncertainty of ±10 °C. The experiments were quenched within 3–5 s below the glass transition temperature (i.e., ~500 °C) by shutting off the electrical power. Pt$_{95}$Rh$_{05}$ capsules with 5.0 mm outer diameter, 4.4 mm inner diameter,

**Fig. 6 | Gold content and Au/Cu ratio of variably evolved potassic vs calc-alkaline melts. a** Melt Au content vs SiO$_2$ content. **b** Melt Au/Cu ratio vs SiO$_2$ content. Red circles and yellow diamonds are potassic melt inclusion data from Sanjiang, China (this study) and Bingham Canyon[26], USA, respectively; blue triangles are calc-alkaline melt inclusion and volcanic glass data from the eastern Manus basin[25,29], Papua New Guinea. The curves show quantitative modeling results based on crystallization experiments on a Sanjiang mafic potassic magma[41] and experimentally constrained D$^{MSS/SM}$ and D$^{SL/SM}$ of Cu and Au (ref. 66), with the solid lines reflecting ideal fractional behavior of the sulfides, and the dashed lines ideal equilibrium behavior. Blue lines reflect fractionation of 100% MSS; green lines fractionation of 100% SL, and purple lines fractionation of 95% MSS + 5% SL. MSS monosulfide solid solution, SL sulfide liquid, SM silicate melt.

fractionation is also indicated by the nature and compositional trends recorded in sulfides found in arc crustal cumulates[44,46,49,50,52–54].

In summary, the results of this study suggest that the close genetic association of Au-rich deposits with potassic magmas is not the result of initial Au enrichment in the magma, but either a consequence of preferential Au extraction by the exsolving magmatic-hydrothermal fluids in the presence of high alkali contents[14], or due to selective Au precipitation from the fluids at the site of ore deposition[16,17]. Based on the fact that potassic magmas in the Sanjiang region and elsewhere did not solely form Au-rich deposits but also Au-poor deposits[41,42,55,56], and that some Au-rich porphyry Cu deposits are also associated with calc-alkaline magmas[1,55,56], we prefer the latter explanation. Potassic magmas commonly occur in extensional tectonic settings[3,15], which likely affected the magma emplacement depth and extent of magma fractionation and thus the characteristics of the related magmatic-

and 7.2 mm or 10 mm length were used to reduce capsule deformation ($Pt_{95}Rh_{05}$ alloy is stiffer than pure Pt) and volatile diffusion ($Pt_{95}Rh_{05}$ alloy recrystallizes much slower). A capsule liner made of gem-quality, single-crystal olivine or zircon was used to minimize loss of Fe and S to the noble metal capsule, e.g., refs. 24,41. The noble metal capsules were sealed using a pulsed PUK-U3 arc welder.

Three different starting materials were prepared by mixing powders (63–160 µm) of (1) the pure spinel lherzolite component in sample ME42 from West Eifel (see Fig. 5), (2) the phlogopite hornblendite vein component in the same sample, and (3) a mantle phlogopite megacryst from West Eifel. The ratio of amphibole to phlogopite in the starting materials was varied to produce partial melts with notably different $K_2O$ contents (Supplementary Data 7). The oxygen fugacity of the experiments was internally buffered by the addition of different amounts of sulfides and sulfates in the starting materials. Thoroughly mixed starting materials were dried at -150 °C for 2 h to remove moisture, and then stored in a desiccator. Modal phase abundances in the run products were estimated based on SEM-EDS single-element maps[41]. Major element compositions of the experimental partial melts were analyzed by EPMA. Vanadium concentrations in the melts and olivine were analyzed by LA-ICP-MS to quantify the experimental $fO_2$, using the approach of Shishkina et al. [60].

## EPMA analysis

A JEOL JXA-8200 electron probe micro-analyzer hosted at BGI was used to analyze (1) spinel inclusions and their olivine host in the natural mafic potassic rocks (Supplementary Data 4), (2) silicate melts in the run products of the 10 mantle partial melting experiments (Supplementary Data 8), (3) two olivine-hosted glassy melt inclusions in the Vulcano basaltic scoria sample (LS01) (Supplementary Data 10), and (4) six quartz-hosted glassy melt inclusions in two Sanjiang rhyolite tuff samples (JC4 and JC5) (Supplementary Data 10). The samples were coated with a -12 nm thick carbon film. For the analysis of silicate melts, the utilized conditions were 15 kV and 10 nA, with a beam defocused to 10–20 µm. Counting times of Na and K on peak and background before and after the peak were 10 and 5 s, respectively; for Fe, Mg, Al, Si, Ca, Ti, P, and Mn they were 20 and 10 s, respectively; and for S, Cl, and ±F they were 60 and 30 s, respectively. Na and K were analyzed first to minimize diffusive loss during the analysis. The elements were standardized on albite (Na, Si), orthoclase (K), metallic Fe, enstatite (Mg), spinel (Al), wollastonite (Ca), $MnTiO_3$ (Ti, Mn), apatite (P), fluorite (F), vanadinite (Cl), and barite and sphalerite (S). Sulfur was measured at a wavelength in the middle between the S peak positions of $BaSO_4$ and ZnS. This approach causes a maximum error of 10%, irrespective of the actual S valence state. Four synthetic, basaltic to andesitic glasses and the NIST SRM610, with known S contents in the range of 0.058 to 0.41 wt%, were analyzed as secondary S standards; the measured S contents agree with the reference values within an uncertainty of ± 10%. For the analysis of spinel and olivine, the conditions were 15 kV and 30 nA, with a beam defocused to 2 µm. Silicon, Ti, Al, Cr, Fe, Mn, Mg, and Ca were measured. Counting times of all the elements on peak and each background were set to 20 and 10 s, respectively. Chromium was calibrated on metallic Cr, whereas the standards for the other elements were the same as those used for silicate melt analysis.

## LA-ICP-MS analysis

Two LA-ICP-MS systems were used to analyze melt inclusions: (1) a GeoLas-Pro 193 nm ArF Excimer laser attached to an Elan DRC-e quadrupole ICP-MS hosted at BGI; (2) the same type of laser attached to an Agilent 7900 quadrupole ICP-MS hosted at the Institute of Geological Sciences, University of Bern, Switzerland. The latter system has a much stronger detection capacity and was thus used to analyze Au and Ag concentrations in large melt inclusions (-32–140 µm in diameter; e.g., Fig. 2). Moreover, analyte signal to background intensity

ratios of S and Cl were also much higher using this instrument. Unexposed, single melt inclusions were drilled out entirely, together with part of the adjacent host mineral, e.g., refs. 61,62. (Fig. 7). The laser systems were operated at 5–10 Hz with an energy density of 3–10 J/cm² on the sample surface. The laser beam size was adjusted in the range of 16–160 µm, depending on the inclusion size, such that it fully covered the inclusion volume while including as little host mineral as possible. The ICP-MS systems were tuned to a ThO rate of 0.10 ± 0.05 % and a rate of doubly-charged $^{42}Ca$ ions of 0.25 ± 0.05 % based on measurements of NIST SRM610 and 612 glasses. The GSE-1G or GSD-1G glass was used as external standard for most elements, whereas a basaltic andesite glass, afghanite or scapolite standard was used for quantifying S and Cl, and the NIST SRM612 glass was used for quantifying Au. Isotopes analyzed at BGI include $^{23}Na$, $^{25}Mg$, $^{27}Al$, $^{30}Si$, $^{31}P$, $^{32}S$, $^{35}Cl$, $^{39}K$, $^{43}Ca$, $^{49}Ti$, $^{51}V$, $^{53}Cr$, $^{55}Mn$, $^{57}Fe$, $^{60}Ni$, $^{65}Cu$, $^{85}Rb$, $^{88}Sr$, $^{89}Y$, $^{90}Zr$, $^{93}Nb$, $^{133}Cs$, $^{137}Ba$, $^{139}La$, $^{163}Dy$, $^{173}Yb$, $^{232}Th$, and $^{238}U$, whereas those analyzed at the Institute of Geological Sciences include $^{23}Na$, $^{25}Mg$, $^{27}Al$, $^{29}Si$, $^{31}P$, $^{34}S$, $^{35}Cl$, $^{39}K$, $^{43}Ca$, $^{49}Ti$, $^{51}V$, $^{53}Cr$, $^{55}Mn$, $^{57}Fe$, $^{65}Cu$, $^{75}As$, $^{85}Rb$, $^{88}Sr$, $^{90}Zr$, $^{93}Nb$, $^{98}Mo$, $^{109}Ag$, $^{137}Ba$, $^{181}Ta$, $^{197}Au$, and $^{208}Pb$. The dwell time was 300 ms for $^{197}Au$, 30 ms for $^{109}Ag$, 20 ms for $^{27}Al$, $^{34}S$, $^{35}Cl$, and $^{43}Ca$, and 10 ms for the remaining isotopes.

Melt inclusion data were quantified by subtracting the contribution of host mineral to the mixed inclusion and host signals, using either a fixed $Al_2O_3$ content as internal standard, or bulk-rock trends of $Al_2O_3$ vs. $SiO_2$ or FeOt vs. $SiO_2$ (ref. 63) (FeOt means total iron expressed as the calculated ferrous amount; Supplementary Data 2 and 9). For example, the $Al_2O_3$ content of glassy melt inclusions determined by EPMA was used as internal standard for olivine-hosted melt inclusions from Vulcano and for quartz-hosted melt inclusions from Sanjiang, whereas the bulk-rock $Al_2O_3$ content was used for melt inclusions in rocks with very low phenocryst contents (e.g., mafic rocks from Feixian, Fangcheng, Shiprock, and WP-STA), and the bulk-rock trend of $Al_2O_3$ vs. $SiO_2$ was used for melt inclusions in rocks with high phenocryst contents (e.g., mafic to intermediate rocks from Two Buttes and Sanjiang). The sum of all major element oxides was normalized to 100 wt%. To achieve optimal host subtraction, the MgO, FeOt, CaO, $P_2O_5$, or $Na_2O$ content of some melt inclusions was fixed at a value[63], based on the composition of bulk rocks or well-quantified, similarly-evolved melt inclusions (Supplementary Data 2 and 9). Sulfur and Cl concentrations in melt inclusions that were analyzed with the Elan DRC-e ICP-MS at BGI were quantified using an in-house afghanite standard that contains 4.4 wt% S and 4.4 wt% Cl, but sophisticated corrections were required in this case[64]. As an alternative approach, sulfur concentrations in the same melt inclusions were also quantified without any correction using a basaltic glass standard B35 that contains 5240 µg/g S (ref. 4). The two approaches returned S concentrations that are in most cases consistent within a relative difference of 5–10% (Supplementary Data 2). Sulfur and Cl concentrations in melt inclusions analyzed by the Agilent 7900 ICP-MS at the Institute of Geological Sciences in Bern were quantified without any correction using scapolite standard Sca17 as external standard that contains 2500 µg/g S and 2.9 wt% Cl. The validity of the results is confirmed by the fact that S and Cl concentrations of olivine- and quartz-hosted glassy melt inclusions analyzed in this way are identical to those analyzed by EPMA (Supplementary Data 2 and 9). Overall, the S and Cl concentrations obtained in the two labs are consistent within the one standard deviation of multiple analyses (Supplementary Data 2). Since the Au concentrations in the analyzed melt inclusions are very low (i.e., only a few ng/g), contamination and interference issues need to be carefully assessed. If possible, the sample surface was cleaned by a few seconds pre-ablation with a large laser beam size before changing to a smaller beam size and reaching the inclusion (Fig. 7). All $^{197}Au^+$ signals (also those obtained from the external NIST SRM612 standard) were corrected for $^{181}Ta^{16}O^+$ interference based on the TaO production rate determined on a synthetic, Ta-rich (1300 µg/g Ta) but Au-free silicate

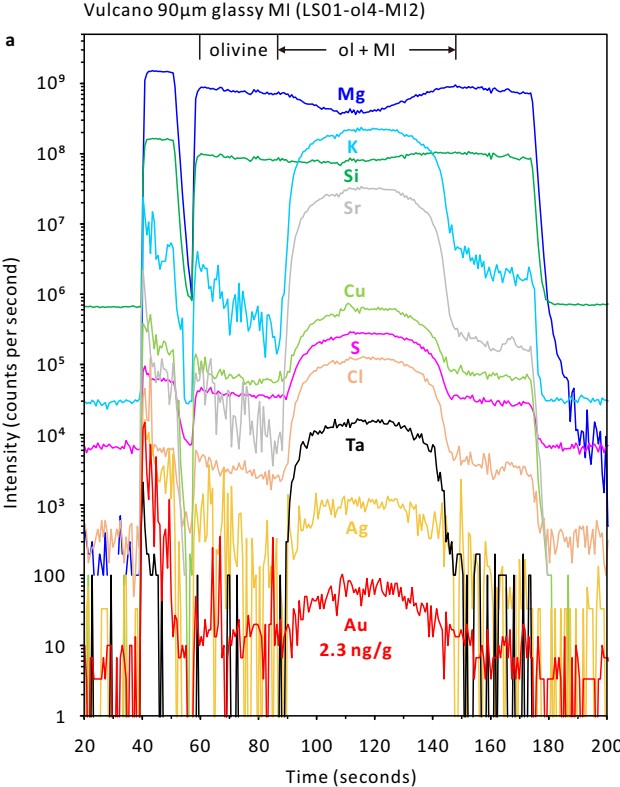

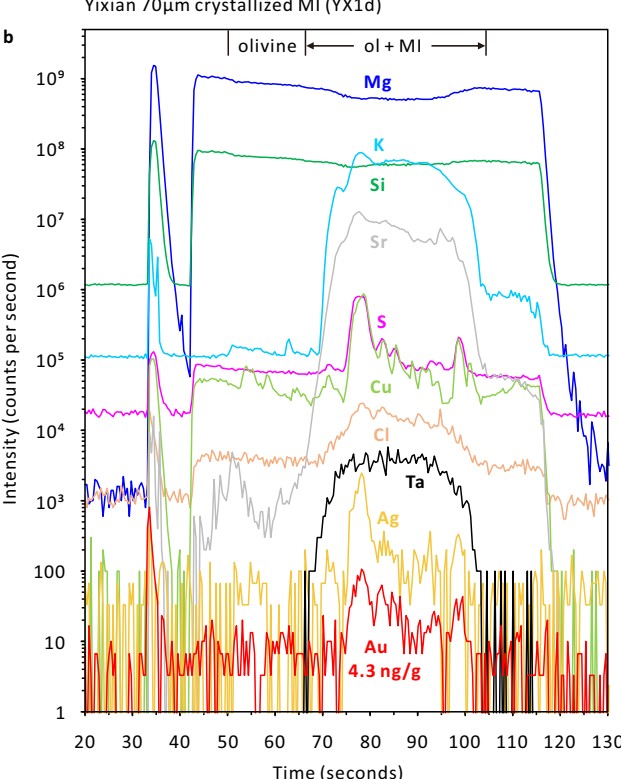

**Fig. 7 | Transient LA-ICP-MS signals obtained from two olivine-hosted melt inclusions.** At the beginning of each analysis, the sample surface was cleaned with a laser beam size exceeding the one that was ultimately used to drill out the inclusion, in order to minimize surface contamination effects. **a** Signal obtained from a glassy melt inclusion from Vulcano, which measured ~90 μm in diameter and contained 9.3 wt% MgO, 2.1 wt% $K_2O$, 46 wt% $SiO_2$, 1,100 μg/g Sr, 130 μg/g Cu, 1,500 μg/g S, 3,200 μg/g Cl, 0.28 μg/g Ta, 0.073 μg/g Ag, and 0.0023 μg/g Au. Based on measurements on a synthetic, Ta-rich and Au-free silicate glass, the Ta-oxide production rate was determined at 0.092%; hence, $^{181}Ta^{16}O^+$ contributed 11% to the total $^{197}Au^+$ signal in this particular melt inclusion. **b** Signal obtained from a crystallized

melt inclusion from Yixian, which measured ~70 μm in diameter and contained 11 wt% MgO, 1.0 wt% $K_2O$, 50 wt% $SiO_2$, 800 μg/g Sr, 71 μg/g Cu, 1,800 μg/g S, 610 μg/g Cl, 0.21 μg/g Ta, 0.065 μg/g Ag, and 0.0043 μg/g Au. The strong correlation of the Au signal with the signals of S, Cu, and Ag suggests that most of the Au was stored in a sulfide daughter crystal. Co-entrapment of sulfide is unlikely because the parental magma was sulfide-undersaturated during olivine crystallization. Note that the shape of the Ta signal is very different from that of Au in this case, which precludes the $^{197}Au^+$ signal to be caused by an $^{181}Ta^{16}O^+$ interference ($^{181}Ta^{16}O^+$ contributed only 5.1% of the total $^{197}Au^+$ signal in this analysis).

glass, which produced an artificial Au signal corresponding to ~0.00092 μg/g Au per μg/g Ta. The different transient shapes of the Au and Ta signals within the inclusion interval obtained from some crystallized melt inclusions clearly demonstrates that the Au signals were not produced by an $^{181}Ta^{16}O^+$ interference (e.g., Fig. 7b). Based on the one standard deviation for the measured inclusion assemblages, the uncertainty of the Au concentrations listed in Supplementary Data 2 and 9 is better than 20–30% (see also ref. 65). Vanadium concentrations in olivine hosts were quantified by normalizing the sum of major element oxides to 100 wt% (Supplementary Data 4).

Sulfide inclusions and experimental phases were analyzed using the Elan DRC-e ICP-MS at BGI. Sulfide inclusions were analyzed by drilling out individual, unexposed inclusions as a whole together with part of the adjacent host. The NIST SRM610 glass was used as external standard for most elements, whereas the Fe/S ratio was quantified using a synthetic pyrrhotite standard (Laflamme Po724 SRM of the memorial University of Newfoundland). Analyzed isotopes include $^{25}Mg$, $^{27}Al$, $^{30}Si$, $^{32}S$, $^{43}Ca$, $^{49}Ti$, $^{55}Mn$, $^{57}Fe$, $^{59}Co$, $^{62}Ni$, $^{65}Cu$, $^{66}Zn$, $^{75}As$, $^{82}Se$, $^{98}Mo$, $^{105}Pd$, $^{107}Ag$, $^{111}Cd$, $^{125}Te$, $^{137}Ba$, $^{197}Au$, $^{205}Tl$, $^{208}Pb$, and $^{209}Bi$. The sulfide inclusion analyses were quantified by subtracting silicate host from the mixed signals until no Si was left, and then normalizing the sum of S, Fe, Cu, Co, and Ni to 100 wt% (Supplementary Data 3 and 6). Palladium concentrations in the sulfide inclusions were corrected for the contribution of the $^{65}Cu^{40}Ar^+$ interference using a correction equation established through the analysis of three variably Cu-rich but

essentially Pd-free sulfides[44] (i.e., bornite, chalcopyrite, and chalcocite).

Vanadium concentrations in the silicate melts and olivine of the experimental products were quantified by normalizing the sum of major element oxides to 100 wt% (Supplementary Data 8). The GSE-1G glass was used as external standard for all elements in this case.

**Numerical modeling**

To understand the observed trends in Au concentration and Au/Cu ratio in the evolving melts during magma differentiation, we ran quantitative models in which different types and proportions of magmatic sulfides crystallized in either ideal fractional or ideal equilibrium mode (Fig. 6). The initial melts in these models contained 52 wt% $SiO_2$, 0.25 wt% S, 100 μg/g Cu, and 2.0 ng/g Au, akin to the most mafic potassic melts at Sanjiang. Because the concentration of S in the residual silicate melts increases during magma differentiation at sulfide-undersaturated conditions, but at the same time the solubility of S decreases due to the change in melt composition and decreasing temperature, the magmas inevitably reached sulfide saturation at 55–60 wt% $SiO_2$ in the residual melt. In these models it is assumed that S precipitates only in the form of sulfides, as no other S-rich phases were observed in the Sanjiang mafic to intermediate potassic rocks. The timing of sulfide saturation and the amounts of precipitated sulfides as a function of magma crystallinity or as a function of the $SiO_2$ content of

the residual melt were constrained based on the following equations:

$$C_{Sulfur}^{SM} = -0.000004292 \times \left(C_{SiO_2}^{SM}\right)^3 + 0.001703 \times \left(C_{SiO_2}^{SM}\right)^2 - 0.1887 \times C_{SiO_2}^{SM} + 6.396 \, (R^2 = 0.90)$$

(1)

$$C_{SiO_2}^{SM} = 0.00002645 \times \left(X^{Total}\right)^3 - 0.00004695 \times \left(X^{Total}\right)^2 + 0.08073 \times X^{Total} + 52.29 \, (R^2 = 0.91)$$

(2)

where $C_{Sulfur}^{SM}$ is the solubility of S in the residual melt, $C_{SiO_2}^{SM}$ is the SiO$_2$ content of the residual melt, $X^{Total}$ is the total magma crystallinity, and the units are in wt%. The amounts of precipitated sulfides were calculated based on a sulfide S content of 36.5 wt%. The amounts of Au and Cu partitioning into minerals other than sulfides were neglected[28]. Previous experimental studies suggest that the partition coefficients of Au and Cu between sulfide and silicate melt change strongly as a function of $f$O$_2$, magma temperature, and the FeOt content of the silicate melt[51,66]. The oxygen fugacity of the modeled magmas was fixed at ΔFMQ + 2.0, a realistic value for variably evolved Sanjiang potassic magmas[41] (Figs. 3d and 4). The magma temperature and the FeOt content of the residual melt were related to the melt SiO$_2$ content based on equations:

$$T = -19.46 \times C_{SiO_2}^{SM} + 2232 \, (R^2 = 0.98)$$

(3)

$$C_{FeOt}^{SM} = 0.003891 \times \left(C_{SiO_2}^{SM}\right)^2 - 0.7929 \times C_{SiO_2}^{SM} + 38.42 \, (R^2 = 0.97)$$

(4)

where $T$ is the magma temperature in °C, and $C_{FeOt}^{SM}$ is the FeOt content of the residual melt in wt%. Using the above equations and experimentally derived empirical equations for metal partitioning between sulfide and silicate melt[66], the bulk mineral–melt partition coefficients of Au and Cu can be expressed as a function of the SiO$_2$ content of the residual melt. The endmember case of ideal equilibrium crystallization of sulfides was then modeled based on the equation:

$$C^{SM}/C_0^{SM} = 1/\left[1 + X/100 \left(D^{Bulk} - 1\right)\right]$$

(5)

whereas the fractional crystallization of sulfides was modeled based on the equation:

$$C^{SM}/C_0^{SM} = (1 - X/100)^{\left(D^{Bulk} - 1\right)}$$

(6)

where $C^{SM}$ is the metal concentration in the residual melt, $C_0^{SM}$ is the metal concentration in the original melt, $X$ is the crystal fraction in wt%, and $D^{Bulk}$ is the bulk partition coefficient of the metal between the crystallized minerals and the residual melt. Equation (1) to (4) are based on crystallization experimental results of a Sanjiang mafic potassic magma[41]. Equations (5) and (6) correspond to equations 4.17 and 4.18, respectively, in Rollinson[67].

## Data availability
The authors declare that all data that were generated in this study are provided in the files Supplementary Data 1–10. The raw data underlying the figures are available in the file Source Data. Source data are provided with this paper.

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

## Acknowledgements

We thank Anna Gioncada for having provided sample material from Vulcano, Zhuang Duan, Wensheng Gao, and Shiguang Du for their help in collecting the samples from China, and Cliff Shaw for helpful information regarding sampling in the West Eifel region. We also like to thank Alex Rother and Raphael Njul for preparing olivine and zircon liners and polishing experimental samples, and Detlef Krauße and Dorothea Wiesner for their assistance with the EPMA and SEM-EDS analysis, respectively. A.A. and J.C. acknowledge funding by the Deutsche Forschungsgemeinschaft (DFG, German Research Foundation; Nr. 440924553). T.P. acknowledges funding by the Swiss National Science Foundation (grant Nr. 206021_170722 to Daniela Rubatto and T.P.). J.C. was additionally supported by the Ministry of Science and Technology of China (Nr. 2023YFF0804200) and the National Natural Science Foundation of China (Nr. 42321001).

## Author contributions

J.C. and A.A. conceived the project and collected the samples. T.P. conducted the LA-ICP-MS analyses of Au concentrations in silicate melt inclusions. A.A. designed the piston cylinder experiments. J.C. performed the petrography, the experiments, all the other LA-ICP-MS and EPMA measurements, the data reduction, and the numerical modeling, and wrote the manuscript with input from A.A. and T.P.

## Competing interests

The authors declare no competing interests.
