## [Peer Review File · Nature Communications]

REVIEWER COMMENTS

Reviewer #1 (Remarks to the Author):

This is a great paper, and I recommend acceptance. It is well written, and the data appear sound. Importantly, they will have significant impact in terms of fundamental understanding of gold ore genesis. These results provide counterpoint to a long standing but unproven belief that magma-related gold ore deposits are intrinsically related to metal enriched magmas.

I think the paper could be improved by including representative photos of melt inclusions; e.g., the ones for which analytical spectra are shown in Fig 2. Also, it would be helpful to know if there was any attempt to acquire or investigate samples from volcanic rocks hosting the Lihir deposit in PNG or Cripple Creek in Colorado, USA as both are well known examples of large hydrothermal gold deposits that are hosted by alkalic mafic rocks. Ok if not, as the rocks maybe altered or the melt inclusions may be poorly developed, preventing useful analysis.

Overall, the data support the conclusions and claims. The methodology appears sound and there is enough data to reproduce the results.

Recommendation: Accept with minor revision (i.e. addition of melt inclusion photos).

Reviewer #2 (Remarks to the Author):

This study introduces the gold content of mafic to felsic potassic magmas and concludes that potassic magmas have not very high gold contents. This manuscript is generally good, but major modifications

are needed to reach the standard of Nature Communications. Below are my comments that might be considered to improve the manuscript:

1. In webber et al., 2013 geology, the average gold content of MORB is 0.34 ppb, which is different from data on Figure 3. Why ?

2. Why did authors choose these 11 rocks? Is them representative of the epithermal gold and gold-rich porphyry-type ore deposits are associated with potassic magmas?

3. What is the detection limit of the Au you tested? Error? What is the internal standard?

4. In Sanjiang region, potassic rocks are not related to hydrothermal gold deposits (such as Ailaoshan gold belt), so they may have low gold contents in nature. Studying these rocks can solve your problem?

5. Although the partition coefficients of Au and Cu between sulfide phases and silicate melt are relatively high, the partition coefficients of Au and Cu are certainly different, MSS-dominated fractionation should cause the enrichment or loss of Au?

6. Au-rich deposits with potassic magmas is not the result of initial Au enrichment in magma, why Au-rich deposits are associated with potassic magmas? If the reason is the preferential Au extraction by the exsolving magmatic-hydrothermal fluids, why do calc-alkaline series not extract Au preferentially?

Response to reviewer comments

Reviewer #1 (Remarks to the Author):

This is a great paper, and I recommend acceptance. It is well written, and the data appear sound. Importantly, they will have significant impact in terms of fundamental understanding of gold ore genesis. These results provide counterpoint to a long standing but unproven belief that magma-related gold ore deposits are intrinsically related to metal enriched magmas.

I think the paper could be improved by including representative photos of melt inclusions; e.g., the ones for which analytical spectra are shown in Fig 2. Also, it would be helpful to know if there was any attempt to acquire or investigate samples from volcanic rocks hosting the Lihir deposit in PNG or Cripple Creek in Colorado, USA as both are well known examples of large hydrothermal gold deposits that are hosted by alkalic mafic rocks. Ok if not, as the rocks maybe altered or the melt inclusions may be poorly developed, preventing useful analysis.

Reply: Representative photographs of melt inclusions have been included (please see our new Fig. 2). We did not have access to samples from the Cripple Creek deposit, and the Lihir deposit is being studied by the group in Geneva (Schirra & Zajacz, 2023, conf.goldschmidt.info/goldschmidt/2023/meetingapp.cgi/Paper/17925). However, our samples from Southwestern China (the Sanjiang region) and Northern China (i.e., Sihetun, Yixian, Feixian, and Fangcheng) are associated with large magmatic-hydrothermal Au deposits (e.g., Chang & Audétat, 2023, doi.org/10.1130/G50502.1; Wang et al., 2020, doi.org/10.1130/G46662.1). Corresponding text has been added in the Methods part (lines 277–280).

Overall, the data support the conclusions and claims. The methodology appears sound and there is enough data to reproduce the results.

Recommendation: Accept with minor revision (i.e. addition of melt inclusion photos).

Reply: Done. Please see above.

Reviewer #2 (Remarks to the Author):

This study introduces the gold content of mafic to felsic potassic magmas and concludes that potassic magmas have not very high gold contents. This manuscript is generally good, but major modifications are needed to reach the standard of Nature Communications. Below are my comments that might be considered to improve the manuscript:

1. In webber et al., 2013 geology, the average gold content of MORB is 0.34 ppb, which is different from data on Figure 3. Why?

Reply: Webber et al. (2013) compiled 22 published data of several studies prior to 2000, which would actually return an average Au content of 1.0 ppb if all data were considered. However, the six highest values (2.3–4.5 ppb) were excluded without clear explanation. Furthermore, half a year before the appearance of the Webber et al. (2013) article, Jenner & O'Neill (2012, doi.org/10.1029/2011GC004009) published a much more extensive data set on Au concentration in 204 MORB glass samples using *in-situ* LA-ICP-MS, which paper was not considered by Webber et al. (2013) (maybe due to too short notice). Anyway, the study of Jenner & O'Neill (2012) is much more comprehensive and their analytical technique is more trustworthy, for which reason we refer to that study.

2. Why did authors choose these 11 rocks? Is them representative of the epithermal gold and gold-rich porphyry-type ore deposits are associated with potassic magmas?

Reply: We chose the studied samples because (1) these samples are widely distributed

over the world, because (2) they cover a wide range of bulk-rock K₂O contents, and (3) because they contain relatively large and well-preserved melt inclusions.

As mentioned above, our samples from the Sanjiang region in Southwestern China and those from Sihetun, Yixian, Feixian, and Fangcheng in Northern China are closely associated with large magmatic-hydrothermal gold deposits (e.g., Chang & Audétat, 2023, doi.org/10.1130/G50502.1; Wang et al., 2020, doi.org/10.1130/G46662.1). To clarify this issue, corresponding text has been added at lines 277–280.

3. What is the detection limit of the Au you tested? Error? What is the internal standard?

Reply: The limits of detection (LODs) were listed in Supplementary Tables 2 and 9 for those melt inclusions in which the Au signal was not significantly higher than the Au signal of the background and of the pure host. These LODs were calculated at the 95% confidence level following Longerich et al. (1996, doi.org/10.1039/ja9961100899) and Pettke et al. (2012, doi.org/10.1016/j.oregeorev.2011.11.001). Most of the melt inclusions that were chosen for Au analysis were actually large enough to give a clear Au signal (e.g., new Fig. 7).

Regarding analytical uncertainty, there exists no reliable method to calculate the error of analyses of melt inclusions that are fully enclosed within other minerals. Halter et al., 2002 (Chemical Geology, doi.org/10.1016/S0009-2541(01)00372-2) discussed this in great detail. Their formulation offers a calculation of the minimum uncertainty associated with an inclusion analysis (their equation 15). As discussed in Pettke et al. (2012, doi.org/10.1016/j.oregeorev.2011.11.001), for example, the most reliable analytical uncertainty is obtained from the measurement of a series of individual inclusions that are compositionally identical, thus representing a measure of the external reproducibility of the method. Pettke et al. (2012) report for the measurement of Au in fluid inclusions a 1SD uncertainty of 10–20% for Au concentrations of ca. 50–100 ng/g. Given the use of a much more sensitive ICP-MS instrument in this work (the sensitivity for Au is about 10 times higher than in the

instrument used in Pettke et al., 2012), the above uncertainty closely corresponds to that expected here for the melt inclusions containing 1–10 ng/g Au. Based on the 1SD uncertainty for the measured melt inclusions of a given sample in the present study, the error of per single measurement should be better than 20–30%. The following text has thus been added in the Methods part (lines 430–433): “Based on the one standard deviation for the measured melt inclusions of a given sample in this study, the uncertainty of the Au concentrations listed in Supplementary Tables 2 and 9 is better than 20–30% (see also ref. 65).”

The selection of the internal standard is clearly explained in the Methods part (lines 390–404).

4. In Sanjiang region, potassic rocks are not related to hydrothermal gold deposits (such as Ailaoshan gold belt), so they may have low gold contents in nature. Studying these rocks can solve your problem?

Reply: The above statement is not correct. Our studied samples are clearly associated with several hydrothermal Au deposits. For example, the giant Beiya skarn-porphyry Au-Cu system, where some of our studied potassic felsic rock samples were collected (i.e., samples BL2, BY4, and BY5#; Supplementary Table 9), is genetically related to a potassic felsic porphyry stock (e.g., He et al., 2015, doi.org/10.2113/econgeo.110.6.1625; Li et al., 2016, doi.org/10.1016/j.oregeorev.2015.05.003; Zhou et al., 2016, doi.org/10.1016/j.oregeorev.2016.06.008; Mao et al., 2017, doi.org/10.1016/j.oregeorev.2017.02.003; Liu et al., 2018, doi.org/10.1016/j.jseaes.2018.04.034; Chang and Audetat, 2023, doi.org/10.1130/G50502.1). Some recent studies argued that even the Ailaoshan lode Au deposits formed by magmatic-hydrothermal fluids associated with the studied Sanjiang potassic magmas (e.g., Wang et al., 2020, doi.org/10.1007/s00126-019-00922-3, and Liu et al., 2015, doi.org/10.1016/j.jseaes.2014.11.006).

5. Although the partition coefficients of Au and Cu between sulfide phases and silicate melt are relatively high, the partition coefficients of Au and Cu are certainly different, MSS-dominated fractionation should cause the enrichment or loss of Au?

Reply: As shown by the quantitative modeling curves in Fig. 6, fractionation of pure MSS would indeed lead to an increase in the Au/Cu ratio in the residual silicate melt, whereas the absolute Au concentration remains \pm constant. The fact that also the Au/Cu ratio remained \pm constant suggests that not only MSS fractionated but also a small amount of SL (~5% if equilibrium fractionation is considered). Text was added to the caption of Fig. 6 to make this clearer.

6. Au-rich deposits with potassic magmas is not the result of initial Au enrichment in magma, why Au-rich deposits are associated with potassic magmas? If the reason is the preferential Au extraction by the exsolving magmatic-hydrothermal fluids, why do calc-alkaline series not extract Au preferentially?

Reply: Two possible reasons were already provided in the introduction: “One option is indicated by an experimental study showing that the partitioning of Au into magmatic fluids is amplified in the presence of high alkali contents (Zajacz et al., 2010).

Alternatively, the commonly extensional tectonic setting in which potassic magmas are generated (Richards, 2009; Loucks, 2012) may promote the development of shallowly emplaced, little evolved magma reservoirs, which, in turn, may affect the hydrothermal evolution and lead to selective Au precipitation in the deposits (Murakami et al., 2010; Koděra et al., 2014).” We now provide a more explicit explanation in the Summary section (lines 232–246).

REVIEWERS' COMMENTS

Reviewer #1 (Remarks to the Author):

The revised manuscript and reply from the authors satisfies the minor concerns expressed in my first review. Thanks for adding the images of the melt inclusions. The paper is acceptable for publication from my perspective.

This ms is significant. It challenges a long standing, poorly substantiated, and largely empirical belief that magmas related to hydrothermal gold deposits, including giant ones, are metal enriched. The data show the opposite. This means that hydrothermal processes play a predominant role in concentrating gold to form ore grade mineralization. The quality of the work and the breadth of samples analyzed are impressive. The documentation provided appear to provide enough detail for checking and reproducing the analytical results.

Reviewer #2 (Remarks to the Author):

I have no further comments and recommend to accept it.